# An Efficient Numerical Simulation for the Fractional COVID-19 Model Using the GRK4M Together with the Fractional FDM

Yasser Ibrahim [1], Mohamed Khader [2,3], Ahmed Megahed [2], Fawzy Abd El-Salam [1] and Mohamed Adel [4,*]

[1] Department of Mathematics, Faculty of Science, Taibah University, Al-Madinah 41411 , Saudi Arabia; yabdelwahab@taibahu.edu.sa (Y.I.); fabdelrahman@taibahu.edu.sa (F.A.E.-S.)
[2] Department of Mathematics, Faculty of Science, Benha University, Benha 13518, Egypt; mmkhader@imamu.edu.sa (M.K.); ahmed.abdelbaqk@fsc.bu.edu.eg (A.M.)
[3] Department of Mathematics and Statistics, College of Science, Imam Mohammad Ibn Saud Islamic University (IMSIU), Riyadh 11566, Saudi Arabia
[4] Department of Mathematics, Faculty of Science, Cairo University, Giza 12613, Egypt
* Correspondence: adel@sci.cu.edu.eg

**Abstract:** In this research, we studied a mathematical model formulated with six fractional differential equations to characterize a COVID-19 outbreak. For the past two years, the disease transmission has been increasing all over the world. We included the considerations of people with infections who were both asymptomatic and symptomatic as well as the fact that an individual who has been exposed is either quarantined or moved to one of the diseased classes, with the chance that a susceptible individual could also migrate to the quarantined class. The suggested model is solved numerically by implementing the generalized Runge–Kutta method of the fourth order (GRK4M). We discuss the stability analysis of the GRK4M as a general study. The acquired findings are compared with those obtained using the fractional finite difference method (FDM), where we used the Grünwald–Letnikov approach to discretize the fractional differentiation operator. The FDM is mostly reliant on correctly converting the suggested model into a system of algebraic equations. By applying the proposed methods, the numerical results reveal that these methods are straightforward to apply and computationally very effective at presenting a numerical simulation of the behavior of all components of the model under study.

**Keywords:** COVID-19; generalized RK4 method; fractional FDM; Grünwald–Letnikov's approach; stability analysis

## 1. Introduction

Many models have been used to explain a wide range of biological phenomena [1,2] with the first appearance of COVID-19 [3,4]. Mathematical models are constructed as a strategy for gaining insight into the pandemic's mode of effect, transmission, spread, prevention, and control, and the impact of preventive measures such as hand-washing with a disinfectant such as hand sanitizer, increasing social distancing, and wearing face masks [5,6]. In [7,8], the authors looked at the early stages of analyzing COVID-19's distribution in Nigeria. However, a more thorough examination of the most recent models as well as a justifiable and satisfactory assessment are required. The proposed model is considered by many researchers to have different forms, i.e., it is different in the number of equations and in the nature of the given derivatives (ordinary/fractional). Additionally, it is solved numerically by using some numerical methods, and the existence, the convergence, and the stability were given through some articles. For more details about the proposed model, see [9–12].

Most of the models described in the abovementioned studies were based on regular derivatives, with some limitations in the order of the involved DEs. To circumvent these constraints, numerous academics have turned to fractional calculus, which is a rapidly growing branch of mathematics. Non-integer or fractional order differential operators are employed in fractional calculus. They have many properties and are important in demonstrating many scientific phenomena and facts with nonlocal dynamical behavior. The fact that these operators' kernels are nonlocal and nonsingular is their primary advantages. Many studies containing fractional and variable order operators were published to describe many daily problems [13,14].

Many numerical and approximate methods are implemented to solve numerically the COVID-19 in different forms; for example, in [15] published in 2021, the authors used the Atangana–Toufik numerical scheme to numerically solve the proposed COVID-19 model and used the least square curve fitting method to obtain the best values for some of the unknown biological parameters involved in the proposed model; in [16], the authors applied the FDM to study the COVID-19 model; and in [17], the authors applied the nonstandard FDM to study the COVID-19 model.

The main aim and the novelty of this work are to extend the COVID-19 system, which is given in [15], to a fractional-order model. To achieve this aim, the numerical solution for the proposed model is obtained by using the GRK4M and the fractional FDM with the approach of Grünwald-Letnikov. The FDM reduces the presented problem to a system of algebraic equations, substantially simplifying it and subsequently optimizing it to obtain the unknown coefficients and, as a result, the solution. We examine the disease-free equilibrium's locally asymptotically-stability, the unique endemic equilibrium point, the locally asymptotically-stability of its unique endemic equilibrium point, and the globally asymptotically-stability of its endemic equilibrium to provide a qualitative analysis of the proposed model. Finally, we give comparative studies. We highlight the main limitation in this study, which is due to the lack of real data, so we study the model from the theoretical side with the help of its corresponding system of ODE.

The following is how the paper is structured: Section 2 contains the model formulation as well as an explanation of the model's parameters. Section 3 contains some preliminary information and notations on the proposed methods. We provide a qualitative examination of the suggested model in Section 4. In Section 5, we numerically solve the proposed problem by using the GRK4M and the fractional FDM and discuss the stability analysis of the GRK4M. In addition, in Section 6, the numerical simulation and the validation of the GRK4M and the fractional FDM of the model are presented. Finally, in Section 7, we present the conclusion and the planned future work.

## 2. Model Construction

This study presents a COVID-19 model that uses fundamental transmission rates. Let $N(t)$ be the whole human population. Susceptible individuals $S(t)$, exposed individuals $E(t)$, asymptotically infected individuals $I_A(t)$, symptomatic infected individuals $I_S(t)$, quarantined persons $Q(t)$, and individuals who have recovered/removed from COVID-19 $R(t)$ are the seven classifications in this population. Taking this into account, the total population is [15]

$$N(t) = S(t) + E(t) + Q(t) + I_A(t) + I_S(t) + R(t).$$

Assume that $\Lambda$ and $\mu$ are the natural human natality and death rates, respectively. $(S)$ becomes infected after contact with $(E)$ or migration to $(Q)$ at a rate of $\beta$. $(E)$ may be the first to be moved to $(Q)$ or they may get sick without displaying any symptoms $(I_A)$ or $(I_S)$ with rates of $\gamma, \sigma$, and $\eta$, respectively. Individuals who have been quarantined (Q) may also be confirmed infected through a test with symptoms $(I_S)$ or without $(I_A)$ with rates $v$ and $\theta$, respectively. The persons in $(I_A)$ may recover at the rate $r_1$ and the persons in $(I_S)$ may recover at the rate of $r_2$ [15].

In the following, some notations will be useful to formulate the studied model:

$\tau$: The number of people who are transferred to quarantine from those who are susceptible;

$\beta$: Contact rate between people who are susceptible and those who are exposed;

$\delta$: Coronavirus-related death rate in the symptomatically infected individual class;

$\gamma$: Rate of transfer of exposed people to quarantine;

$\eta$: Rate of transfer of persons from the exposed to the symptomatically infected classes;

$\theta$: The proportion of quarantined people who are asymptomatically infected;

$\mu$: Natural death rate;

$v$: Rate of transfer quarantined people to the symptomatically infected people class;

$\sigma$: Rate of exposed people to asymptomatically infected people class;

$\Lambda$: Recruitment (natality) rate;

$r_1$: Asymptomatically infected people's recovery rate;

$r_2$: Symptomatically infected people's recovery rate.

Each of these groups may drop as a consequence of natural death $\mu$, while ($I_S$) may decrease as a result of disease death at the rate of $\delta$. The mortality rate as a result of the disease is not regarded in $I_A$. This model does not account for the risk of reinfection following recovery.

The schematic diagram in Figure 1 is used to build a system, which is discussed below:

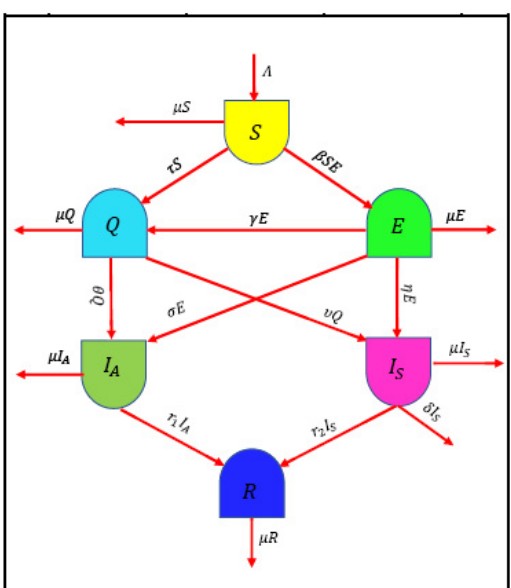

**Figure 1.** Flowchart describing the dynamics of propagation of the COVID-19 model.

$$\frac{dS(t)}{dt} = \Lambda - (\tau + \mu)S(t) - \beta S(t)E(t),$$
$$\frac{dE(t)}{dt} = \beta S(t)E(t) - (\gamma + \mu + \eta + \sigma)E(t),$$
$$\frac{dQ(t)}{dt} = \tau S(t) + \gamma E(t) - (\mu + v + \theta)Q(t),$$
$$\frac{dI_A(t)}{dt} = \sigma E(t) + \theta Q(t) - (\mu + r_1)I_A(t),$$
$$\frac{dI_S(t)}{dt} = \eta E(t) + vQ(t) - (\delta + \mu + r_2)I_S(t),$$
$$\frac{dR(t)}{dt} = r_1 I_A(t) + r_2 I_S(t) - \mu R(t).$$

(1)

A system of fractional differential equations (FDEs) is offered as a mathematical model to characterize the dynamics of propagation for COVID-19 ($\forall\ t \geq 0$):

$$
\begin{aligned}
D^\alpha S(t) &= \Lambda - (\tau + \mu)S(t) - \beta S(t)E(t),\\
D^\alpha E(t) &= \beta S(t)E(t) - (\gamma + \mu + \eta + \sigma)E(t),\\
D^\alpha Q(t) &= \tau S(t) + \gamma E(t) - (\mu + v + \theta)Q(t),\\
D^\alpha I_A(t) &= \sigma E(t) + \theta Q(t) - (\mu + r_1)I_A(t),\\
D^\alpha I_S(t) &= \eta E(t) + vQ(t) - (\delta + \mu + r_2)I_S(t),\\
D^\alpha R(t) &= r_1 I_A(t) + r_2 I_S(t) - \mu R(t),
\end{aligned}
\tag{2}
$$

where $\alpha$ refers to the order of the Liouville–Caputo fractional derivative.

Figure 1 shows the employed coefficients (parameters) in the model together. Additionally, the initial conditions are as follows:

$$
S(0) = S_0, \quad E(0) = E_0, \quad Q(0) = Q_0, \quad I_A(0) = IA_0, \quad I_S(0) = IS_0, \quad R(0) = R_0. \tag{3}
$$

With this model (2) in its fractional form, we can more precisely characterize the effect of the spread of the pandemic of COVID-19 in the future and in history based on the memory effect of fractional derivatives. As it is also known, although the mathematical models with integer derivatives play an important role and have their significance to understand the dynamics of epidemiological systems, they have some limitations, as these systems do not have memory or non-local effects; therefore, these models in the usual form are sometimes inappropriate, and thus, it is necessary to convert several epidemiological models into FDEs to have an effective ability to explore many natural phenomena, facts that have non-local dynamic behavior. As a general case, FDEs are frequently used in the study of anomalous phenomena in nature and in the theory of complex systems as well as when taking into account the properties of the curve over a large extent. Finally, it can explain the time delay, fractal properties, etc. For more details about this model, see [18,19].

## 3. Preliminaries and Notations

In this part, we go over some of the fundamental concepts and useful properties that are used in this work.

**Definition 1.** *The Liouville–Caputo and the Riemann–Liouville derivatives of order $\alpha$ are, respectively, defined by [20,21]:*

$$
D^\alpha f(t) = \frac{1}{\Gamma(m-\alpha)} \int_0^t \frac{f^{(m)}(\xi)}{(t-\xi)^{\alpha-m+1}} d\xi, \qquad m-1 \leq \alpha \leq m, \quad m \in \mathbb{N}.
$$

$$
D_R^\alpha f(t) = \frac{1}{\Gamma(m-\alpha)} \frac{d^m}{dt^m} \int_0^t \frac{f(\xi)}{(t-\xi)^{\alpha-m+1}} d\xi, \qquad m-1 \leq \alpha \leq m, \quad m \in \mathbb{N}.
$$

**Definition 2.** *The shifted Grünwald–Letnikov fractional derivative is defined as follows [22]:*

$$
\frac{d^\alpha y(t)}{dt^\alpha} = \lim_{h \longrightarrow 0} \frac{1}{h^\alpha} \sum_{i=0}^{[\frac{t}{h}]+1} (-1)^i \binom{\alpha}{i} y(t - (i-1)h), \tag{4}
$$

*for some constant h.*

**Lemma 1.** *For $|\ t\ | < \rho$, with arbitrary constant, assume that $\rho y(t)$ can be written as a power series. The Grünwald–Letnikov formula holds for each $0 < \alpha < \rho$ and a series of step size h with $\frac{\tau}{h} \in \mathbb{N}$,*

$$
D_R^\alpha y(\tau) = \frac{1}{h^\alpha} \Delta_h^\alpha y(nh) + O(h), \qquad (h \longrightarrow 0), \tag{5}
$$

where $D_R^\alpha$ refers to the Riemann–Liouville definition of the fractional derivative and

$$\Delta_h^\alpha y(nh) = \sum_{i=0}^{n} (-1)^i \binom{\alpha}{i} y(t_{n-i}).$$

Therefore, by using the relation between both definitions (Liouville–Caputo and the Riemann–Liouville) [23] for $0 < \alpha \leq 1$, we can find the following in the case of the Liouville–Caputo sense:

$$D^\alpha y(\tau) = \frac{1}{h^\alpha} \Delta_h^\alpha y(nh) - \frac{\tau^{-\alpha}}{\Gamma(1-\alpha)} y(0) + O(h), \qquad (h \longrightarrow 0). \tag{6}$$

## 4. Qualitative Analysis of the Proposed Model

As we can see in [24], one of the most important parameters in studying the infectious disease model is the basic reproduction number $\mathfrak{R}_0 = 1$, which is defined and derived by the following formula [15,24]:

$$\mathfrak{R}_0 = \frac{\beta \Lambda}{(\gamma + \mu + \eta + \sigma)(\tau + \mu)} > 0.$$

Additionally, in this section, we present the invariant region of (1).

### 4.1. Invariant Region

It is very important to note that $S(t), E(t), Q(t), I_A(t), I_S(t), R(t)$ are nonnegative for all $t \geq 0$ in the model (1). For all $t \geq 0$, solution via initial positive data remains bounded and positive. From system (1), we can see that $\frac{dN(t)}{dt} = \Lambda - \mu N(t) - \delta I_S(t)$ and $\sup_{t \to +\infty} N(t) \leqslant \frac{\Lambda}{\mu}$. The system (1) will be studied in the feasible region:

$$\Omega = \left\{ (S(t), E(t), Q(t), I_A(t), I_S(t), R(t)) \in \mathbb{R}_+^6 : \quad 0 \leqslant N(t) \leqslant \frac{\Lambda}{\mu} \right\}. \tag{7}$$

Equation (7) is now positive invariant in relation to (1), which means that (1) is epidemiologically well-posed.

The disease-free equilibrium (DFE) is obtained by letting $E = Q = I_A = I_S = R = 0$. Therefore, the equations in (1) indicate that this point is given by the following:

$$\bar{\mathbf{E}}_0 = (S_0, 0, 0, 0, 0, 0) = \left( \frac{\Lambda}{\tau + \mu}, 0, 0, 0, 0, 0 \right).$$

### 4.2. Existence of Endemic Equilibrium Point

Let us denote the endemic equilibrium by $\bar{\mathbf{E}}_1 = (S^*, E^*, Q^*, I_A^*, I_S^*, R^*)$, where the values of all the components $S^*, E^*, Q^*, I_A^*, I_S^*, R^*$ can be found in [15].

**Theorem 1.** $\bar{E}_0$ is locally asymptotically stable if $\mathfrak{R}_0 < 1$.

**Theorem 2.** The system (1) has a unique $\bar{E}_1$.

**Theorem 3.** $\bar{E}_1$ is locally asymptotically stable if $\mathfrak{R}_0 > 1$.

**Theorem 4.** The system (1) has no periodic orbits.

**Theorem 5.** $\bar{E}_1$ for (1) is globally asymptotically stable whenever $\mathfrak{R}_0 > 1$.

The definition of $\bar{\mathbf{E}}_1$ and the proof of the five Theorems 1–5 can be given in [15].

## 5. Numerical Studies

In this section, we use GRK4M and fractional FDM to solve the suggested model numerically. In addition, as part of general research, we cover the stability analysis of the GRK4M.

### 5.1. Procedure of Solution by the GRK4M

To show how to implement GRK4M in solving the proposed model (2), we consider the following general form of the FDE [25]:

$$
\begin{aligned}
D^\alpha u(t) &= \psi(t; u(t)), \quad 0 < \alpha \le 1, \quad 0 < t \le T, \\
u(0) &= u_0.
\end{aligned}
\tag{8}
$$

Now to construct the numerical scheme of the GRK4M [26], we divide the interval $[0, T]$ to $n$ of equal subintervals with the nodes $\{t_0, t_1, \ldots, t_n\}$, where $t_0 = 0$, $t_j = jh$, $j = 1, 2, \ldots n$, and $t_n = T$, and $h = \frac{T}{n}$ is the step size.

The numerical scheme of the GRK4M takes the following form:

$$
u_{n+1} = u_n + \frac{1}{6}(K_1 + 2K_2 + 2K_3 + K_4),
$$

$$
K_1 = \hbar\, \psi(t_n, u_n), \qquad\qquad K_2 = \hbar\, \psi\left(t_n + \frac{1}{2}\hbar, u_n + \frac{1}{2}K_1\right),
\tag{9}
$$

$$
K_3 = \hbar\, \psi\left(t_n + \frac{1}{2}\hbar, u_n + \frac{1}{2}K_2\right), \qquad K_4 = \hbar\, \psi(t_n + \hbar, u_n + K_3),
$$

where $\hbar = \frac{h^\alpha}{\Gamma(\alpha+1)}$.

To study the stability of the GRK4M, we take for simplicity the function $\psi(t; u(t)) = \zeta\, u(t)$, for some constant $\zeta < 0$.

**Theorem 6.** *The numerical scheme* (9) *of the FDE* (8) *with the assumption* $\psi(t; u(t)) = \zeta\, u(t)$, $\zeta < 0$ *is conditionally stable under the following criterion:*

$$
0 < h^\alpha < 12\left|\frac{\Gamma(\alpha+1)}{\zeta}\right|.
\tag{10}
$$

**Proof.** We can rewrite Equation (8) by using GRK4M as follows:

$$
u(t_{j+1}) = u(t_j) + \frac{1}{6}\frac{h^\alpha \zeta}{\Gamma(\alpha+1)} u(t_j), \qquad j = 0, 1, \ldots, n-1.
$$

When the terms of this equation are regrouped, we can obtain the following recurrence formula [27]:

$$
u(t_{j+1}) = \left(1 + \frac{1}{6}\frac{h^\alpha \zeta}{\Gamma(\alpha+1)}\right)^j u_0, \qquad j = 0, 1, \ldots, n-1.
$$

Then, the stability condition of this recurrence formula is given as follows:

$$
\left|\left(1 + \frac{1}{6}\frac{h^\alpha \zeta}{\Gamma(\alpha+1)}\right)\right| < 1,
$$

under some simplifications, we can obtain the condition (10), and this ends the proof. $\square$

*5.2. Procedure of Solution by the Fractional FDM*

First, for system (2), we consider the following:

$$
\begin{aligned}
D^\alpha S_n(t) &= \Lambda - (\tau + \mu)S_n(t) - \beta S_n(t)\, E(_n t),\\
D^\alpha E_n(t) &= \beta S_n(t) E_n(t) - (\gamma + \mu + \eta + \sigma)E_n(t),\\
D^\alpha Q_n(t) &= \tau S_n(t) + \gamma E_n(t) - (\mu + v + \theta)Q_n(t),\\
D^\alpha I A_n(t) &= \sigma E_n(t) + \theta Q_n(t) - (\mu + r_1)I A_n(t),\\
D^\alpha I S_n(t) &= \eta E_n(t) + v Q_n(t) - (\delta + \mu + r_2)I S_n(t),\\
D^\alpha R_n(t) &= r_1 I A_n(t) + r_2 I S_n(t) - \mu R_n(t).
\end{aligned}
\tag{11}
$$

Second, we use $t_n = nh$, where $n = 0, 1, ..., M, Mh = T$ and the abbreviations $S_n$, $E_n$, $Q_n$, $IA_n$, $IS_n$, and $R_n$ for approximation of the true solutions $S(t_n)$, $E(t_n)$, $Q(t_n)$, $IA(t_n)$, $IS(t_n)$, and $R(t_n)$, in the grid point $t_n$.

By applying the definition of the approximate fractional derivative (6)–(11), we obtain the following:

$$
\frac{1}{h^\alpha}\sum_{i=0}^{n+1}(-1)^i\,\psi_i^\alpha\,S_{n+1-i} - \frac{(nh)^{-\alpha}}{\Gamma(1-\alpha)}S_0 = \Lambda - (\tau + \mu)S_n - \beta S_n E_n,
\tag{12}
$$

$$
\frac{1}{h^\alpha}\sum_{i=0}^{n+1}(-1)^i\,\psi_i^\alpha\,E_{n+1-i} - \frac{(nh)^{-\alpha}}{\Gamma(1-\alpha)}E_0 = \beta S_n E_n - (\gamma + \mu + \eta + \sigma)E_n,
\tag{13}
$$

$$
\frac{1}{h^\alpha}\sum_{i=0}^{n+1}(-1)^i\,\psi_i^\alpha\,Q_{n+1-i} - \frac{(nh)^{-\alpha}}{\Gamma(1-\alpha)}Q_0 = \tau S_n + \gamma E_n - (\mu + v + \theta)Q_n,
\tag{14}
$$

$$
\frac{1}{h^\alpha}\sum_{i=0}^{n+1}(-1)^i\,\psi_i^\alpha\,IA_{n+1-i} - \frac{(nh)^{-\alpha}}{\Gamma(1-\alpha)}IA_0 = \sigma E_n + \theta Q_n - (\mu + r_1)IA_n,
\tag{15}
$$

$$
\frac{1}{h^\alpha}\sum_{i=0}^{n+1}(-1)^i\,\psi_i^\alpha\,IS_{n+1-i} - \frac{(nh)^{-\alpha}}{\Gamma(1-\alpha)}IS_0 = \eta E_n + v Q_n - (\delta + \mu + r_2)IS_n,
\tag{16}
$$

$$
\frac{1}{h^\alpha}\sum_{i=0}^{n+1}(-1)^i\,\theta_i^\alpha\,R_{n+1-i} - \frac{(nh)^{-\alpha}}{\Gamma(1-\alpha)}R_0 = r_1 IA_n + r_2 IS_n - \mu R_n,
\tag{17}
$$

where $\psi_i^\alpha = \begin{pmatrix} \alpha \\ i \end{pmatrix}$.

We express the problem defined by (12)–(17) as a constrained optimization problem with the following cost functions (CFs):

$$
CF1 = \frac{1}{h^\alpha}\sum_{i=0}^{n+1}(-1)^i\,\psi_i^\alpha\,S_{n+1-i} - \frac{(nh)^{-\alpha}}{\Gamma(1-\alpha)}S_0 - \Lambda + (\tau + \mu)S_n + \beta S_n E_n,
\tag{18}
$$

$$
CF2 = \frac{1}{h^\alpha}\sum_{i=0}^{n+1}(-1)^i\,\psi_i^\alpha\,E_{n+1-i} - \frac{(nh)^{-\alpha}}{\Gamma(1-\alpha)}E_0 - \beta S_n E_n + (\gamma + \mu + \eta + \sigma)E_n,
\tag{19}
$$

$$
CF3 = \frac{1}{h^\alpha}\sum_{i=0}^{n+1}(-1)^i\,\psi_i^\alpha\,Q_{n+1-i} - \frac{(nh)^{-\alpha}}{\Gamma(1-\alpha)}Q_0 - \tau S_n - \gamma E_n + (\mu + v + \theta)Q_n,
\tag{20}
$$

$$
CF4 = \frac{1}{h^\alpha}\sum_{i=0}^{n+1}(-1)^i\,\psi_i^\alpha\,IA_{n+1-i} - \frac{(nh)^{-\alpha}}{\Gamma(1-\alpha)}IA_0 - \sigma E_n - \theta Q_n + (\mu + r_1)IA_n,
\tag{21}
$$

$$
CF5 = \frac{1}{h^\alpha}\sum_{i=0}^{n+1}(-1)^i\,\psi_i^\alpha\,IS_{n+1-i} - \frac{(nh)^{-\alpha}}{\Gamma(1-\alpha)}IS_0 - \eta E_n - v Q_n + (\delta + \mu + r_2)IS_n,
\tag{22}
$$

$$CF6 = \frac{1}{h^\alpha} \sum_{i=0}^{n+1} (-1)^i \, \psi_i^\alpha \, R_{n+1-i} - \frac{(nh)^{-\alpha}}{\Gamma(1-\alpha)} R_0 - r_1 IA_n - r_2 IS_n + \mu R_n. \tag{23}$$

## 6. Numerical Simulation

### 6.1. Validation of the GRK4M

The goal of this subsection is to demonstrate the accuracy and reliability of the numerical values acquired through the GRK4M. Clearly, the comparison is carried out for the values of the numerical solution in the interval $(0, 140)$ by using the GRK4M with the standard RK4M (SRK4M) at the order of the fractional derivative $\alpha = 1$ and $h = 0.05$, with $S_0 = 0.2$, $E_0 = Q_0 = 0.1$, $IA_0 = IS_0 = 0.2$, $R_0 = 0.0$, where in this case, $\mathfrak{R}_0 = 0.359916 < 1$. This verification is suggested in Figure 2, which shows that the current results are in excellent accord with those obtained by the SRK4M. This also confirms that the proposed method (GRK4M) is good implemented. In addition, the numerical scheme by the introduced method, and the given models (1) and (2) are stable.

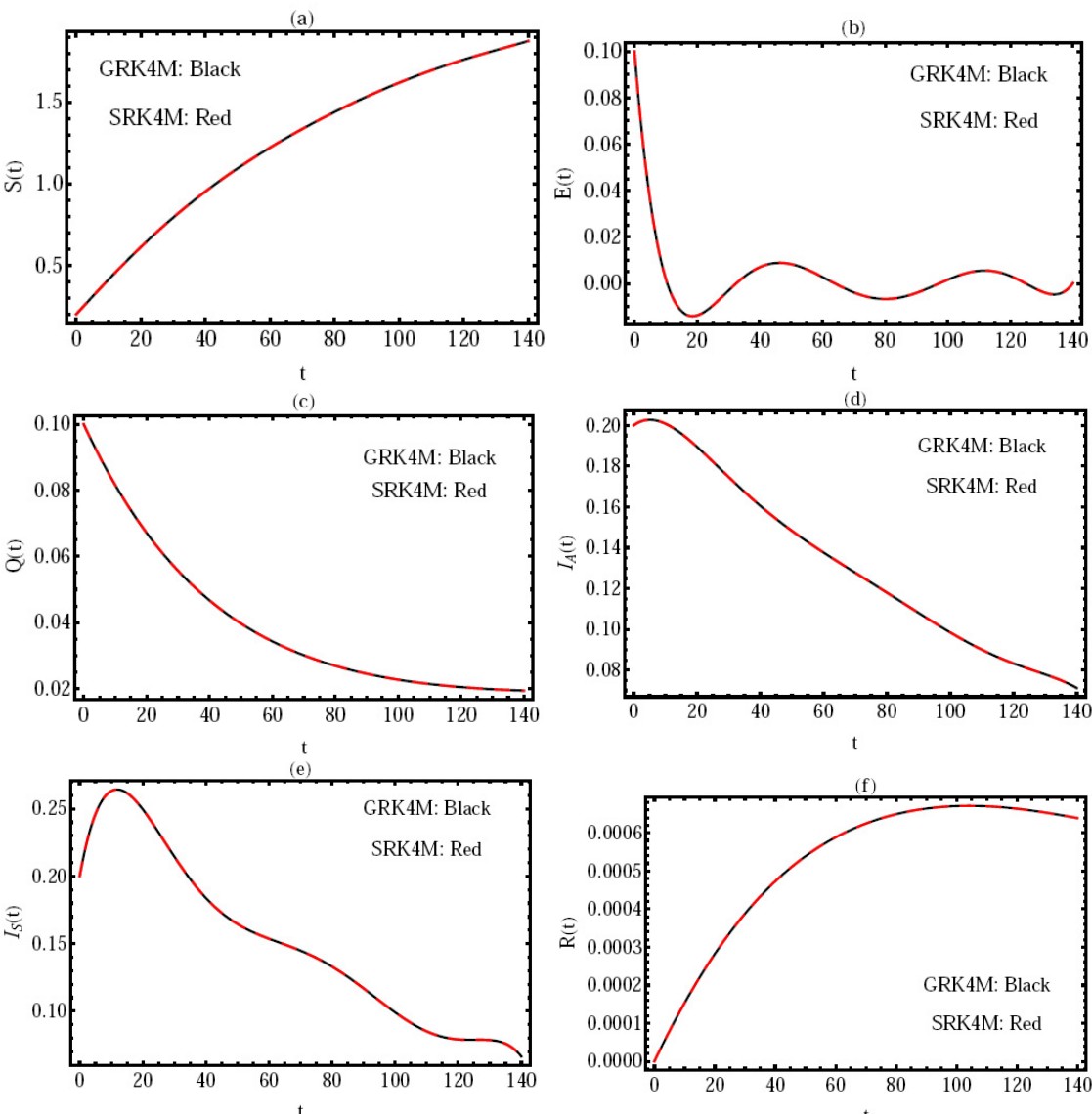

**Figure 2.** The solution by GRK4M and SRK4M at $\alpha = 1$ (**a–f**).

We consider the systems (1) and (2) with the following parameters [15]:

$$\tau = 0.0002, \quad \beta = 0.0805, \quad \delta = 0.000016728, \quad \gamma = 0.00020138,$$

$$\eta = 0.4478, \quad \theta = 0.0101, \quad mu = 0.0106, \quad v = 0.0003208,$$

$$\sigma = 0.0668, \quad \Lambda = 0.02537, \quad r1 = 0.00005734, \quad r2 = 0.00001673.$$

### 6.2. Numerical Results Using GRK4M and the Fractional FDM

We consider (2) with the following parameters [15]:

$$\tau = 0.0002, \beta = 0.08, \delta = 0.000016, \gamma = 0.0002, \eta = 0.44, \theta = 0.01,$$

$$\mu = 0.01, v = 0.0003, \sigma = 0.06, \Lambda = 0.025, \quad r_1 = 0.000057, \quad r_2 = 0.000016,$$

and the different values of $S_0$, $E_0$, $Q_0$, $IA_0$, $IS_0$, $R_0$, and distinct values of the order of the fractional derivative $\alpha$.

In Figure 3, the solution for all components of the model by using GRK4M via distinct values of $\alpha = 1.0, 0.92, 0.82, 0.72$, is presented in the interval $(0, 60)$ and $h = 0.01$, with $S_0 = 0.5$, $E_0 = 0.2$, $Q_0 = 0.1$, $IA_0 = 0.2$, $IS_0 = 0.1$, $R_0 = 0.0$, where in this case, $\mathfrak{R}_0 = 0.359916 < 1$, and in the view of Theorem 1, we can note that $\xi_0$ is locally asymptotically stable.

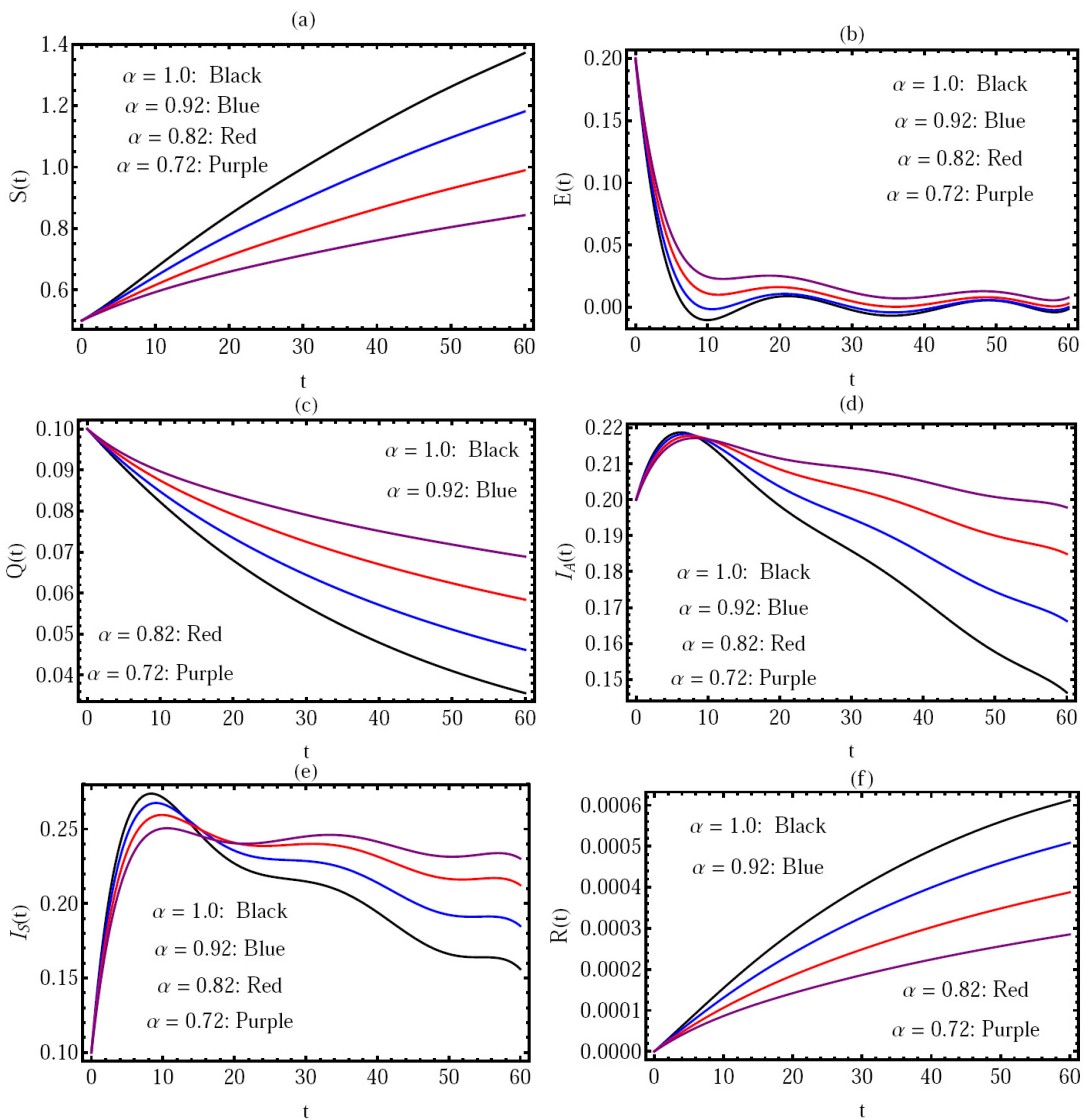

**Figure 3.** The solution by GRK4M via different values of $\alpha$ (**a–f**).

In Figure 4, the solution for all components of the model is presented using the fractional FDM via the same values in Figure 3. From the Figures 3a,b and 4a,b, we can observe that if $\alpha$ decreases, the amounts $S(t)$ and $R(t)$, respectively, will be reduced. Hence, a small memory of the infection effect maximizes the number of COVID-19 healthy individuals. In Figure 5, the solution via distinct values of the initial conditions with $h = 0.05$, $\alpha = 0.95$ is presented in the interval $(0, 120)$, where the components of solution $S(t)$, $E(t)$, $Q(t)$, $I_A(t)$, $I_S(t)$, $R(t)$ are presented in Figure 5a–f, respectively. There are three cases:

i. $S_0 = 0.5$, $E_0 = 0.1$, $Q_0 = 0.1$, $IA_0 = 0.3$, $IS_0 = 0.3$, $R_0 = 0.0$;
ii. $S_0 = 0.3$, $E_0 = 0.2$, $Q_0 = 0.2$, $IA_0 = 0.1$, $IS_0 = 0.2$, $R_0 = 0.0$;
iii. $S_0 = 0.4$, $E_0 = 0.3$, $Q_0 = 0.3$, $IA_0 = 0.2$, $IS_0 = 0.1$, $R_0 = 0.0$.

We note that, in all these cases, the basic reproduction number $\mathfrak{R}_0 < 1$.

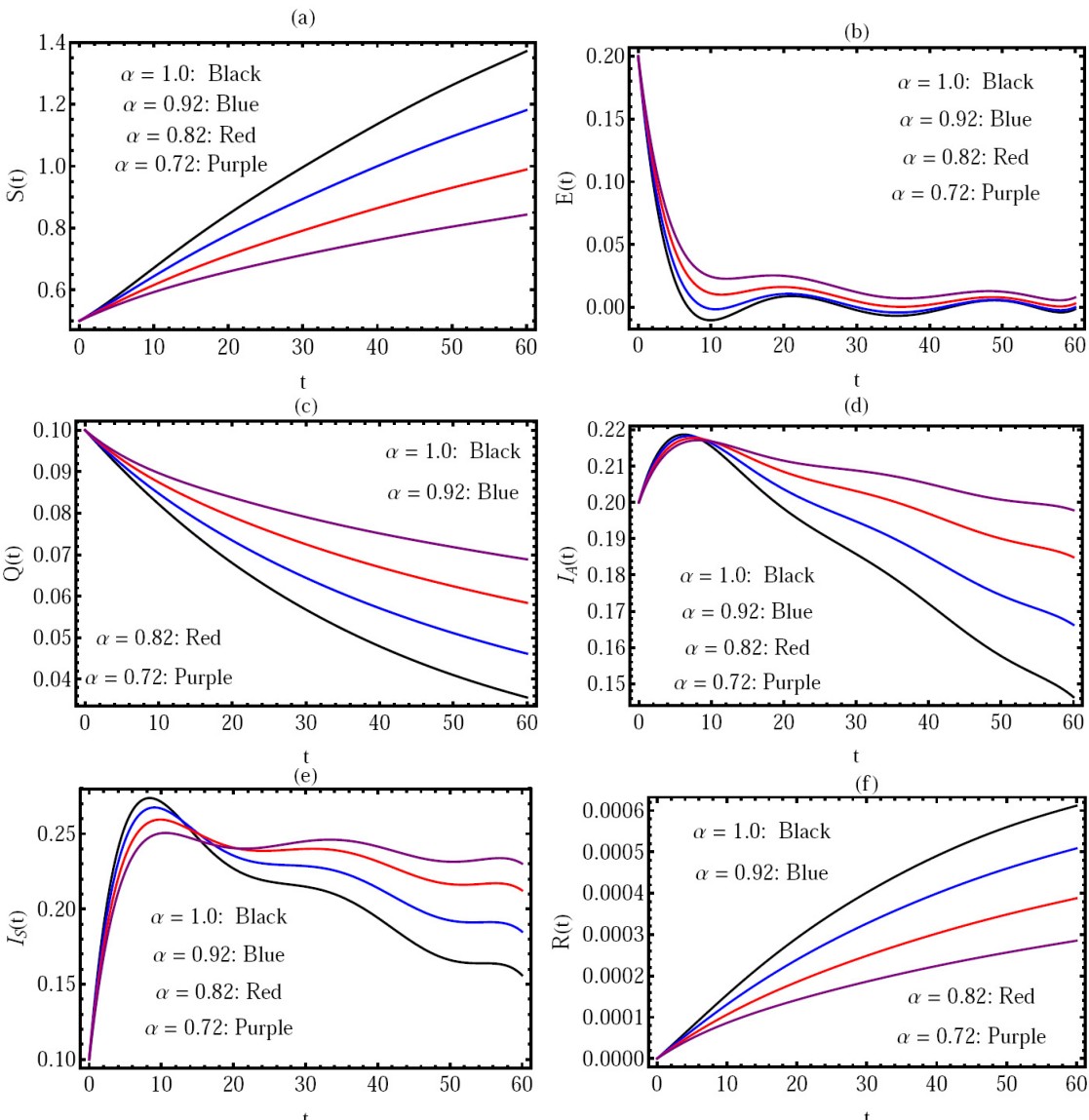

**Figure 4.** The solution by FDM via different values of $\alpha$ (**a–f**).

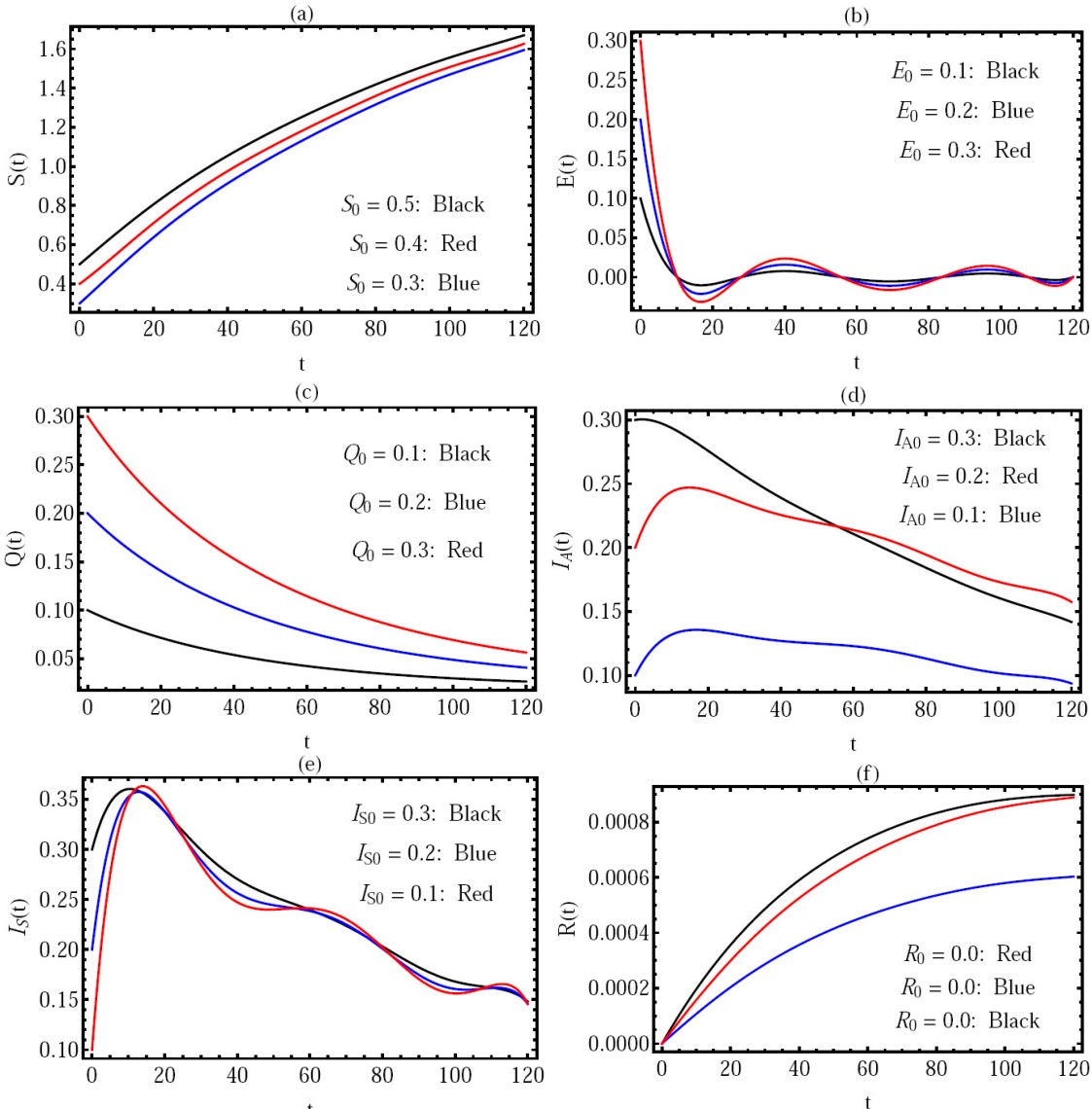

**Figure 5.** The solution via different values of initial values (**a**–**f**).

Finally, in Figure 6, we presented a comparison between the numerical solution obtained by the GRK4M and the fractional FDM at $\alpha = 0.94$ and $h = 0.05$. With the same parameters included in Figure 4, and the following initial conditions:

$$S_0 = 0.2, \quad E_0 = 0.1, \quad Q_0 = 0.1, \quad IA_0 = 0.2, \quad IS_0 = 0.2, \quad R_0 = 0.0.$$

From the COVID-19 dynamics in this figure, we can observe and confirm that the majority of the population recover because the recovered population grows significantly; see Figure 6f. In addition, we can observe and confirm that the populations of those infected and exposed decrease significantly; see Figure 6b,d,e. This means that the majority of the population will be recovered, which results in a decrease in the deaths caused by COVID-19.

We can confirm that the expected behavior of the disease has occurred, and this presents a clear simulation of the model. Additionally, the good physical interpretation of these numerical results is devoted to knowing the behavior of all of the components (stats) of the model with different values of the derivative, not only with $\alpha = 1$.

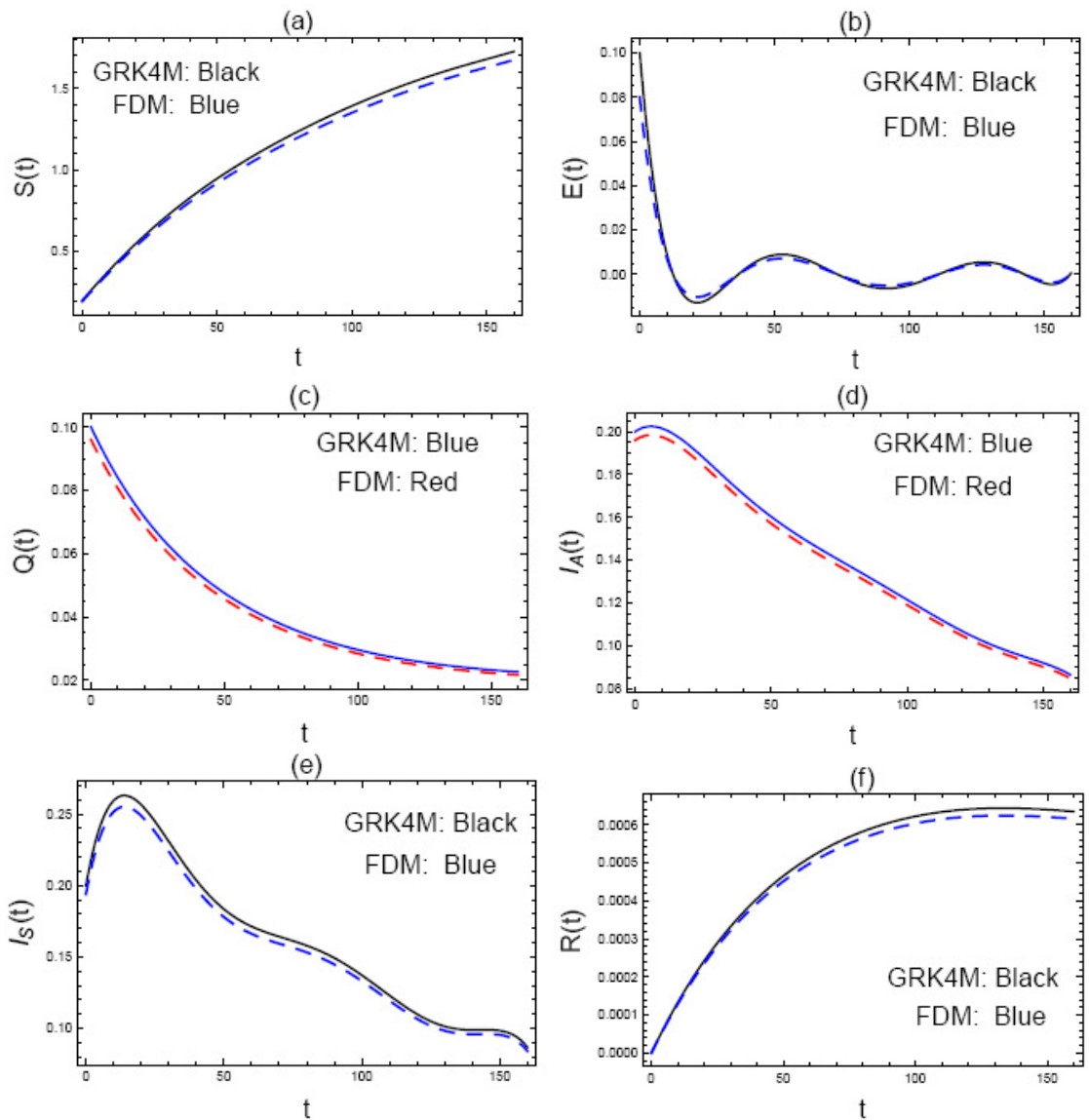

**Figure 6.** Comparison of the solution obtained by GRK4M and FDM at $\alpha = 0.94$ (**a–f**).

## 7. Conclusions and Remarks

The studied problem regarding the COVID-19 model is solved numerically by employing the GRK4M and the fractional FDM with the help of the Grünwald–Letnikov's formula of the fractional derivative. The Caputo-derivative was used here since it simply requires the initial circumstances to be expressed in terms of integer-order derivatives. The proposed model's qualitative analysis is demonstrated by studying the locally asymptotically stable equilibrium of its disease-free equilibrium, the globally asymptotically stable equilibrium of its endemic equilibrium, and the locally asymptotically stable equilibrium of its disease-free equilibrium. Two equilibrium points, the disease-free equilibrium point $\bar{\mathbf{E}}_0$ and the endemic equilibrium point $\bar{\mathbf{E}}_1$, were found for the studied model. The stability analysis shows that $\bar{\mathbf{E}}_0$ is locally asymptotically stable whenever $\mathfrak{R}_0 < 1$ and $\bar{\mathbf{E}}_1$ is globally asymptotically stable whenever $\mathfrak{R}_0 > 1$. Using the fitted values of the parameters showing the spread of the disease, a sensitivity analysis was performed on the parameters in the $\mathfrak{R}_0$ as well as the profile of each state variable. The contact rates between susceptible individuals and transmission rates from the exposure to the symptomatically infected classes are key parameters of the $\mathfrak{R}_0$.

Besides computing numerical solutions with different values of $\alpha$ and $m$, the initial conditions, and the REF, we perform a verification of the proposed approach. We can

conclude that the Caputo-operator is a very efficient tool for investigating the numerical solution for the considered model in our study. The method presents an efficient way of investigating the numerical solution for such a model. Additionally, from the presented comparison, we can see the excellent agreement between our numerical solution for the fractional model obtained by using the GRK4M and the numerical solution obtained using the fractional FDM at various values of $\alpha$. In contrast, the numerical analysis of our work provides field meanings. We performed our numerical analysis using Mathematica. Finally, in the future, the authors are interested extending the model with studies of its optimal control and considering different forms to cover all of the affected parameters of the disease as well as the concerned fractional COVID-19 model can also be analyzed.

In addition, we can recommend that, by increasing the awareness of the population and by adhering to the government's actions, the amount of susceptible individuals is reduced and the number of infected population is increased, and this in turn plays a major role in reducing the severity of infection. As a final conclusion, as we know that most numerical methods have some limitations, the two proposed methods have some shortness since we cannot evaluate the solution at any point but only at the nodes of the given domain, and they are not unconditionally stable. We plan to achieve some future work such as the following:

1. Solving the same model but with different techniques such as the finite element method, the finite volume method, and others;
2. Optimal control of the resulting solutions;
3. Some deepness studies in the theoretical part to describe the COVID-19 model;
4. Change the sense of the fractional derivative to be, for example, Atangana–Baleanu–Caputo or to be variable-order.

**Author Contributions:** Data curation, Y.I.; Formal analysis, A.M. and M.A.; Methodology, M.K. and M.A.; Project administration, Y.I.; Resources, F.A.E.-S.; Software, M.K.; Writing—original draft, A.M. and F.A.E.-S.; Writing review and editing, M.K. and M.A. All authors have read and agreed to the published version of the manuscript.

**Funding:** The manuscript is funded by the Saudi Arabian Ministry of Education's Deanship for Research & Innovation.

**Institutional Review Board Statement:** Not applicable.

**Informed Consent Statement:** Not applicable.

**Data Availability Statement:** All data and material are available for everyone.

**Acknowledgments:** The authors extend their appreciation to the Deputyship for Research & Innovation, Ministry of Education in Saudi Arabia for funding this research work the project number (208/442.) Also, the authors would like to extend their appreciation to Taibah University for its supervision support.

**Conflicts of Interest:** The authors declare no conflict of interest.

## Abbreviations

The following abbreviations are used in this manuscript:

| | |
|---|---|
| COVID-19 | Coronavirus Disease 2019 |
| FDM | finite difference method |
| FDE | fractional differential equation |
| DFE | disease-free equilibrium |
| GRK4M | generalized fourth-order Runge–Kutta method |

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
