# Peer review of "An Efficient Numerical Simulation for the Fractional COVID-19 Model Using the GRK4M Together with the Fractional FDM"

_fractalfract, doi:10.3390/fractalfract6060304_

Round 1

Reviewer 1 Report

Paper deals with an important task. It has a scientific novelty and logical structure. Paper is technically sound. The proposed approach is logical, results are clear.

Suggestions:

  1. Introduction section should be extended using more clearly motivation of this paper.
  2. The mathematical model (2) was obtained. However, it is not specified:
    • the physical content of the alpha parameter;
    • what are the advantages of this model compared to the traditional case (1);
    • why the fractional order derivative in the form (6) was chosen?
  1. Numerical solutions of the obtained mathematical models (1), (2) have been found, but there is no model validation section.
  2. Figure 5 shows the value of the residual error function. However, it is not clear on the basis of which ratios these results were obtained.
  3. Conclusion section should be extended using:
    • limitations of the proposed approach;
    • prospects for the future research.

Author Response

Dear Editor, Greetings

      We would like to thank you for your cooperation and patience. In addition, we appreciate the efforts and valuable comments of the discerning reviewers. We submit a revised version for our manuscript to fix all the issues raised by reviewers.

Please, find below my detailed response to the raised comments of the first Reviewer, point by point.

======================================

Reviewer #1:

[1] Introduction section should be extended using more clearly motivation of this paper.

Response: Done. We added a new paragraph concerning the proposed model and the implemented methods to solve it numerically. Please, see page 3, lines 19-26 in the revised version of the paper.

[2] The mathematical model (2) was obtained. However, it is not specified:

      (i) the physical content of the alpha parameter;

Response: Here the parameter $\alpha$ refers to the order of the Liouville-Caputo fractional derivative.

     (ii) what are the advantages of this model compared to the traditional case (1);

Response: We added a new paragraph (after the system (2)) concerning the advantages of the proposed model in its fractional form.

      (iii) why the fractional order derivative in the form (6) was chosen?

Response: Firstly, we apology for this confusion. We modified this part of the paper to avoid this confusion. We should note that the Grunwald-Letnikov formula (5) holds for the Riemann-Liouville definition, but we used the Liouville-Caputo sense so we implement the relation between both definitions to obtain the form (6).

[3] Numerical solutions of the obtained mathematical models (1), (2) have been found, but

     there is no model validation section.

Response: Thank you for this good remark. We added a subsection with model validation by

solving the system (2) by using the proposed method (GRK4M) and the standard RK4M with \alpha=1. Please, see subsection 6.1 in the revised version.

[4] Figure 5 shows the value of the residual error function. However, it is not clear on the

      basis of which ratios these results were obtained.

Response: We apology for this mistake. We deleted this figure and were satisfied with figure 5 (in the new version of the paper) to achieve the comparison between the two implemented methods. The residual error function was calculated by implement another analytical and approximate method (Spectral method) in another situation.

[5] Conclusion section should be extended using:

     (i) limitations of the proposed approach;

     (ii) prospects for the future research.

Response: Done. Please, see the conclusion section of the revised version of the paper.

Finally, we revised all the paper again and corrected some of grammatical error.

Reviewer 2 Report

Comments on the manuscript titled

“An efficient numerical simulation for a fractional Covid-19 model by using the GRK4M together with and the fractional FDM ”

Manuscript Number:  fractalfract-1718860 

After roughly reading your manuscript, I felt that the novelty in your manuscript was very limited. For example, the theory based on a fractional Covid-19 model was not new; the numerical method based on the fourth-order Runge-Kutta method and finite difference method for solving a fractional Covid-19 model was nothing new.

In my opinion,Reject.

Author Response

Dear Editor, Greetings

      We would like to thank you for your cooperation and patience. In addition, we appreciate the efforts and valuable comments of the discerning reviewers. We submit a revised version for our manuscript to fix all the issues raised by reviewers.

Please, find below my detailed response to the raised comments of the second Reviewer.

======================================

Reviewer #2:

    After roughly reading your manuscript, I felt that the novelty in your manuscript was very limited. For example, the theory based on a fractional Covid-19 model was not new; the numerical method based on the fourth-order Runge-Kutta method and finite difference method for solving a fractional Covid-19 model was nothing new.

Response:

     There are many techniques in the field of numerical analysis like the finite difference method (FDM), the finite element method, the finite volume method, the spectral collocation methods (SCM), and many other techniques. Also, the number of models and systems that were solved (and still solving) by one of these mentioned techniques is large. The novelty of how to apply a certain technique to solve a certain system that was NOT solved using this technique before and present some theoretical studies (stability, convergence, error analysis) resulting from the application of this technique to solve the given model and the happiness holds when the obtained results are convergence, stable, etc. Also, the novelty holds if you compared the results obtained from two different techniques in the same manuscript, in general, there are an infinite number of novelty attitudes. On the other hand, in the last two years, there are many research papers solving Covid-19 models using some well-known methods like FDM, SCM, and others, for example:

  1. In Ref.[17] published in 2021, the authors used the Atangana-Toufik numerical scheme to solve numerically the proposed the Covid-19 model and used the least square curve fitting method to obtain the best values for some of the unknown biological parameters involved in the proposed model.
  2. In this paper published in 2021: http://dx.doi.org/10.31764/jtam.v5i1.3468, the authors applied the finite difference method to study the Covid-19 model.
  3. In this paper published in 2022: https://pubmed.ncbi.nlm.nih.gov/35135201/, the authors applied the nonstandard finite difference method to study the Covid-19 model.

 In all these published works (and there are many others), the models are well-known and also the techniques are known, which confirms our reply. In our study, we solved the proposed Covid-19 model using two different techniques and compared the results obtained.

To improve the manuscript, the following modifications are done:

[1] We added a new paragraph concerning the proposed model and the implemented methods to solve it numerically. Please, see page 3, lines 19-26 in the revised version of the paper.

[2] We added a new paragraph (after the system (2)) concerning the advantages of the proposed model in its fractional form.

[3] We added a subsection with model validation by solving the system (2) by using the proposed method (GRK4M) and the standard RK4M with \alpha=1. Please, see subsection 6.1 in the revised version.

[4] We modified some parts in the numerical results section (in the new version of the paper) to achieve the comparison between the two implemented methods. The residual error function was calculated by implement another analytical and approximate method (Spectral method) in another situation.

[5] We added the future research in the conclusion section.

[6] Finally, we revised all the papers again and corrected some of the grammatical errors.

Round 2

Reviewer 1 Report

All comments were taken into account by the authors.

Reviewer 2 Report

The author has done his best to revise the manuscript.